*Report*

EMBO
Molecular Medicine

# Plasma Proteome Profiling to detect and avoid sample-related biases in biomarker studies

Philipp E Geyer[1,2] (iD), Eugenia Voytik[1], Peter V Treit[1], Sophia Doll[1,2], Alisa Kleinhempel[3], Lili Niu[2], Johannes B Müller[1], Marie-Luise Buchholtz[3], Jakob M Bader[1], Daniel Teupser[3], Lesca M Holdt[3] & Matthias Mann[1,2,*] (iD)

## Abstract

Plasma and serum are rich sources of information regarding an individual's health state, and protein tests inform medical decision making. Despite major investments, few new biomarkers have reached the clinic. Mass spectrometry (MS)-based proteomics now allows highly specific and quantitative readout of the plasma proteome. Here, we employ Plasma Proteome Profiling to define quality marker panels to assess plasma samples and the likelihood that suggested biomarkers are instead artifacts related to sample handling and processing. We acquire deep reference proteomes of erythrocytes, platelets, plasma, and whole blood of 20 individuals (> 6,000 proteins), and compare serum and plasma proteomes. Based on spike-in experiments, we determine sample quality-associated proteins, many of which have been reported as biomarker candidates as revealed by a comprehensive literature survey. We provide sample preparation guidelines and an online resource (www.plasmaproteomeprofiling.org) to assess overall sample-related bias in clinical studies and to prevent costly miss-assignment of biomarker candidates.

**Keywords** biomarker discovery; mass spectrometry; plasma proteomics; sample quality; study design

**Subject Categories** Biomarkers; Proteomics

## Introduction

Protein levels determined in blood-based laboratory tests can be useful proxies of diseases. These biomarkers assess normal physiological status, pathogenic processes, or a response to an exposure or intervention (FDA-NIH:Biomarker-Working-Group, 2016). Proteins and enzymes constitute the largest proportion of laboratory tests, reflecting the importance of the plasma proteome in clinical diagnostics (Geyer *et al*, 2017). Typical protein biomarkers such as the enzymes aspartate aminotransferase (ASAT) and alanine aminotransferase (ALAT) for the diagnosis of liver diseases or cardiac troponins indicating myocardial necrosis are used routinely in clinical decision making. Enzymatic activity or antibody-based laboratory tests are performed in high-throughput and at relatively low costs, as the standard of health care. However, specific biomarkers are only available for a very limited number of conditions and most have been introduced decades ago (Anderson *et al*, 2013). There is thus a critical need to make the biomarker discovery process more efficient.

Protein-binder assays quantifying many plasma proteins in parallel have become available (Gold *et al*, 2010; Assarsson *et al*, 2014), resulting in large-scale biomarker mining efforts (Ganz *et al*, 2016; Herder *et al*, 2018; Sun *et al*, 2018). Orthogonal to those technologies, mass spectrometry (MS)-based proteomics has become increasingly powerful in all domains of protein research (Aebersold & Mann, 2003, 2016; Munoz & Heck, 2014). MS measures the mass and fragmentation spectra of tryptic peptides derived from the sample with very high accuracy. Because these peptide and fragment masses are unique, MS-based proteomics is inherently specific, which can be an advantage over enzyme tests and immunoassays (Wild, 2013). Within its limit of detection, MS-based proteomics can analyze all proteins in a system and is unbiased and hypothesis-free in this sense.

The proteomic community has developed guidelines for the development, specificity, and potential clinical application of biomarkers. These discuss quality standards and emphasize the importance of selecting cohorts that are appropriate in size, thus ensuring the statistical significance of potential findings (Mischak *et al*, 2010; Surinova *et al*, 2011; Skates *et al*, 2013; Hoofnagle *et al*, 2016; Geyer *et al*, 2017). That being said, there are no systematic procedures in place to assess the proteome-wide effects of pre-analytical handling of blood-based samples. Considering that plasma samples are often collected during daily clinical routine and variably processed, sample collection and processing clearly have the potential to negatively influence clinical studies, making it difficult to uncover true biomarkers, while potentially contributing incorrect ones. Especially in case–control studies, any difference in the

1 Department of Proteomics and Signal Transduction, Max Planck Institute of Biochemistry, Martinsried, Germany
2 NNF Center for Protein Research, Faculty of Health Sciences, University of Copenhagen, Copenhagen, Denmark
3 Institute of Laboratory Medicine, University Hospital, LMU Munich, Munich, Germany
   *Corresponding author. Tel: +49 89 8578 2557; E-mail: mmann@biochem.mpg.de

collection and processing of samples may result in systematic bias. So far, relatively little attention has been paid to this crucial aspect on a proteome-wide scale and these studies mainly investigate pre-analytical effects (Rai *et al*, 2005; Timms *et al*, 2007; Schrohl *et al*, 2008; Qundos *et al*, 2013; Hassis *et al*, 2015).

Recently, we developed "Plasma Proteome Profiling", an automated MS-based pipeline for high-throughput screening of plasma samples (Geyer *et al*, 2016a). In this article, we apply this technology to systematically assess the quality of individual samples and clinical studies with the aim to identify generally applicable quality marker panels. Blood collection and subsequent errors in preparation are likely sources of plasma contamination. To address this issue, we construct proteomic catalogs of contaminating cell types as well as proteomic changes that may be induced during processing. This results in three panels of contaminating proteins, recommendations for assessing the quality of plasma samples and for consistent sample processing. We develop an online tool for biomarker studies and test the applicability of the panels on a recent investigation on the effects of weight loss on the plasma proteome (Geyer *et al*, 2016b). A comprehensive literature review of plasma proteome studies highlights that about half of them potentially suffer from limitations related to sample processing.

## Results

### Erythrocyte and platelet proteins in the plasma proteome

During the development of our Plasma Proteome Profiling pipeline and its optimization for high-throughput screening of human cohorts (Geyer *et al*, 2016a), we repeatedly observed proteins that tended to emerge as groups of statistically significant outliers but appeared to be independent of the particular study. We hypothesized that they reflected sample quality issues. Manual and bioinformatic inspection revealed three classes of origin: erythrocytes, platelets, and the blood coagulation system. Consequently, we designed experiments to systematically characterize these main quality issues of the plasma proteome.

First, we acquired reference proteomes of erythrocytes and platelets, which are by far the most abundant cellular components ($5 \times 10^6$ and $3 \times 10^5$ cells per μl). We harvested these cellular components from 10 healthy females and 10 males to obtain representative erythrocytes, platelets, and pure (platelet-free) plasma and further collected platelet-rich plasma and whole blood (Fig 1A; see Materials and Methods). Cell counting confirmed the purity of the samples (Table EV1). All five blood fractions were separately prepared for each individual by our automated proteomic sample preparation pipeline, followed by liquid chromatography coupled to high-resolution mass spectrometry (LC-MS/MS). To create reference proteomes, we generated a very deep library from pooled samples by analyzing extensively pre-fractionated peptides (Kulak *et al*, 2017; see Materials and Methods). A total of 6,130 different proteins were identified from 61,654 sequence-unique peptides (Fig 1B and C). The platelet proteome was the most extensive (5,793 proteins), whereas we detected 2,069 proteins in erythrocytes, 1,682 in platelet-rich plasma, and 912 in platelet-free plasma. The comparison of platelet-rich plasma to platelet-free plasma (84% additional

proteins) demonstrates the extent of proteins that can be introduced by platelets.

Next, we investigated purified samples for all 20 study participants individually. The average numbers of identified proteins and peptides were very consistent in all individuals (Appendix Fig S1). To construct panels of easily detectable and robust quality markers, we calculated the average protein intensities and the coefficient of variation (CV) across the study participants. As a prerequisite, we required that the proteins should be substantially more abundant in erythrocytes as well as platelets rather than in plasma. According to these criteria, we selected the 30 most abundant proteins with CVs below 30% and at least a 10-fold higher expression level in the contaminating cell type than in plasma (Fig 1D and E). NIF3-like protein 1 (NIF3L1), a low-abundance erythrocyte-specific protein, was excluded, because it was inconsistently identified as was the platelet-bound coagulation factor F13A1, whose function makes it an unsuitable platelet marker. The remaining proteins represent our cellular quality marker panels (Table EV2). They overlap by just two proteins (actin/ACTB and glyceraldehyde-3-phosphate dehydrogenase/GAPDH), and their quantities were not correlated with each other (Appendix Fig S2). Thus, they are specific and independent indicators for the origin of plasma quality.

Comparing median expression values of proteins shared between the blood components revealed that plasma proteins do correlate with whole blood (Pearson's correlation coefficient $R = 0.43$), as expected. In contrast, there was no correlation between the platelet, erythrocyte, and plasma proteomes (Appendix Fig S2). This indicates that the levels of cellular proteins in plasma are not a constant fraction of those in the cellular proteomes. The platelet panel was enriched in platelet-rich plasma compared to normal (platelet-free) plasma. Both panels are de-enriched in pure plasma compared to whole blood, however, this effected the erythrocyte panel even more strongly, because centrifugation removes erythrocytes more efficiently than platelets. A histogram of both panels over the abundance range visualizes their distribution in the different blood compartments (Appendix Fig S2). Erythrocytes are 10-fold more abundant and fourfold larger than platelets, and indeed, the corresponding panel proteins have a 42-fold difference in whole blood. In plasma, however, their ratio was nearly one to one, again pinpointing a more efficient removal of erythrocytes than of platelets in standard sample preparation. The fact that several proteins of both panels were still detectable in pure plasma indicates a baseline level of contaminants due to imperfect de-enrichment or the life cycle of these cells. The four most abundant erythrocyte proteins, HBA1, HBB, CA1, and HBD, were present in pure plasma of almost all individuals, whereas lower abundant proteins were only sporadically identified. In contrast, platelet proteins were quantified over a larger abundance range and some of them were found in every individual.

In addition to the sum of panel protein abundances, we calculated their correlation to the standard reference panel defined by the 20 participants to several hundred plasma samples of a previous study (Geyer *et al*, 2016b). A distinct contamination of erythrocyte proteins seems to be a part of the plasma proteome as the erythrocyte panel has in general a relatively high correlation between the reference cohort erythrocyte levels and the plasma samples in the above-mentioned study. In contrast, in many plasma samples there

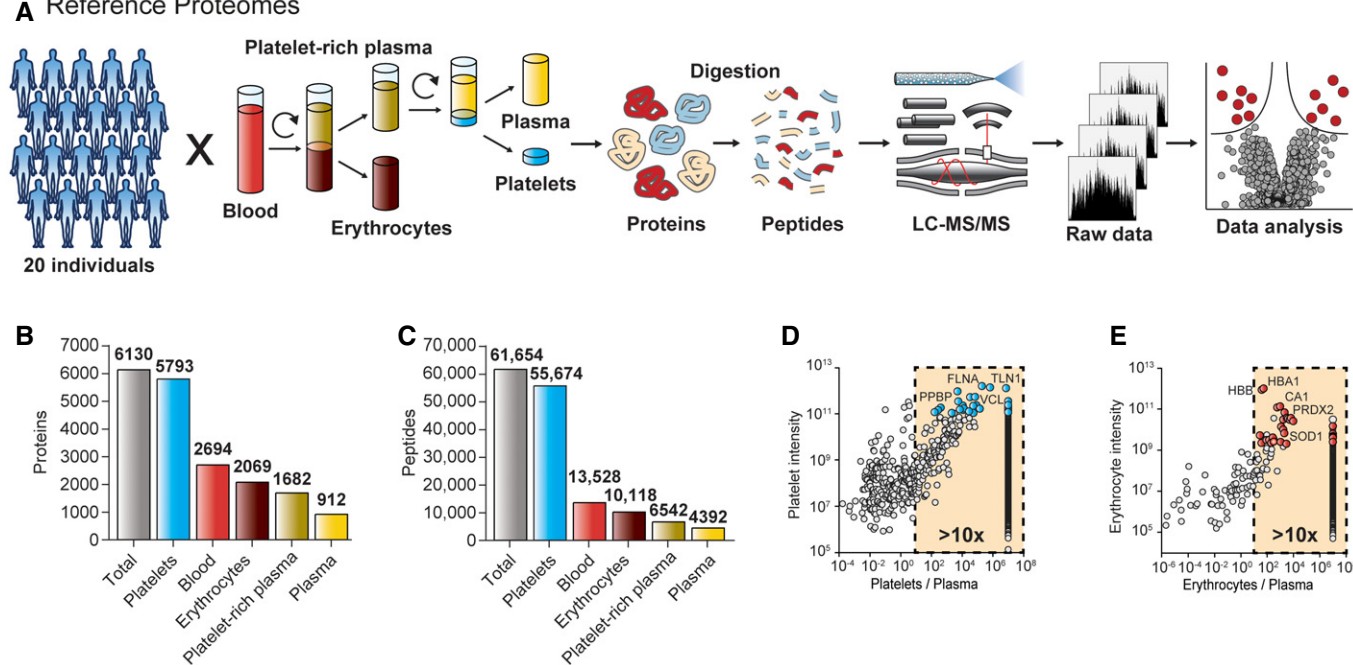

**Figure 1. Identification of blood cell markers.**

A   Study outline and proteomic workflow. Erythrocytes, thrombocytes, platelet-rich, and platelet-free plasma were generated from 10 healthy female and male individuals by differential centrifugation and successive purification steps. To generate reference proteomes for each of the blood compartments, the respective protein samples of the 20 study participates were digested to peptides.

B, C   Proteins (B) and peptides (C) identified for platelets, erythrocytes, platelet-rich, and platelet-free plasma.

D, E   Selection of the most suitable quality marker proteins for (D) platelet contamination (blue dots) and (E) erythrocyte contamination (red dots) based on their abundance, the platelet/erythrocyte-to-plasma ratio, and the coefficient of variation. Proteins that were only detected in platelets or erythrocytes, but not in plasma are aligned on the right side of the graph.

was no correlation detectable between the reference cohort platelet levels and the plasma samples in the study. In practice, a correlation > 0.5 indicated that the proteins are present as a result of contamination (Appendix Fig S3A–C). Note that an apparent contaminant protein could still be applied as a biomarker—however, in this case its abundance value should be different from the pattern in the reference quality panel.

## Serial dilution experiments validate the erythrocyte and platelet quality marker panels

To determine whether the two protein panels correctly quantify contamination in plasma, we generated four pools of erythrocytes and platelets from five study participants at a time. These pools were diluted in nine steps into platelet-free plasma for a total range of $10^7$, followed by cell counting and proteomic analysis (Fig 2A). This resulted in an expected decrease in the cellular proteome ratio to plasma (Fig 2B and C). All but two of the panel proteins were consistently quantified over the dilution range. As the protein within each panel has the same origin, we defined a single variable for each cell type by summing their intensities and dividing by the summed intensities of all quantified plasma proteins. This yielded two remarkably robust "contamination indices" that turned out to be linear with respect to the cell numbers determined by cell cytometry (Table EV3; $R = 0.98$ and 0.99, Fig 2D and E). Spiked-in

contaminations of 1:100 could readily be detected, which corresponds to a concentration of 70,000 erythrocytes or 30,000 platelets per µl plasma.

## Quality marker panel for blood coagulation

In addition to contamination due to cellular constituents, partial and variable coagulation could contribute to systematic bias in biomarker studies. Indeed, we had found coagulation-related proteins to be connected to sample handling from finger pricks while developing our plasma proteomics pipeline (Geyer *et al*, 2016a). In clinical practice, an anticoagulant is pre-added to commercially available containers so that it is combined with blood upon withdrawal. Prompt inversion mixes the anticoagulant with the blood, yielding pure plasma after centrifugation (Fig 3A). Any delay in adding or mixing could cause partial coagulation—in the extreme case of missing anticoagulant and waiting for 30 min, one would obtain serum instead of plasma.

To generate a panel for assessing blood coagulation, we systematically compared 72 plasma vs. 72 serum samples (four individuals, 18 aliquots). From a total of 2,099 quantified proteins, 299 were significantly altered (Fig 3B). The most significantly de-enriched proteins after clotting were typical constituents of the coagulation cascade such as fibrinogen chains alpha (FGA), beta (FGB), and gamma (FGG) ($P < 10^{-130}$, > 40-fold), whereas the platelet-associated

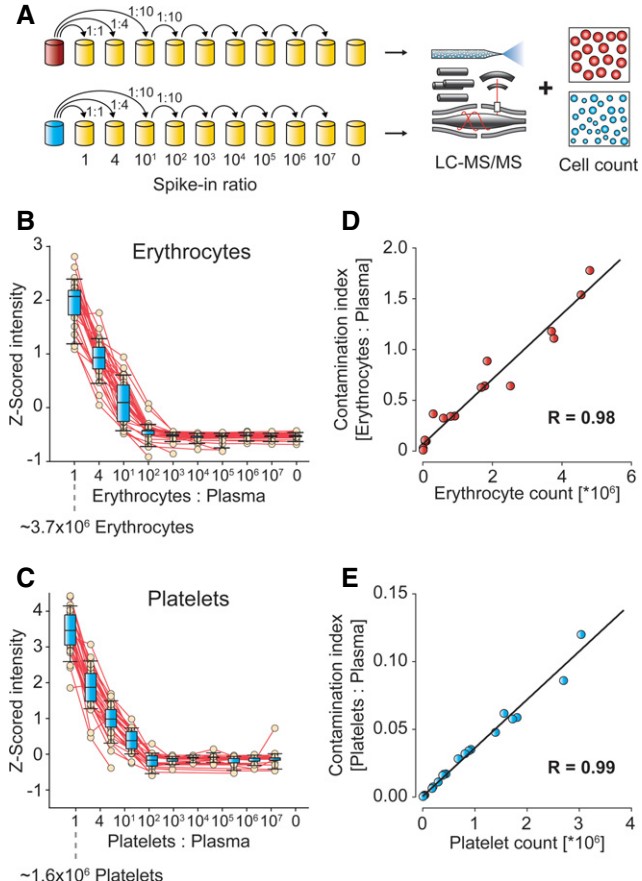

**Figure 2. Spike-in of erythrocyte and platelet fractions into pure plasma.**

A     Dilution and analysis scheme.
B, C  Protein intensities were Z-scored across the dilution series (B) for the 29 quality markers of the erythrocyte panel and (C) for the 29 markers of the platelet panel as a function of their spike-in proportion to plasma. Whiskers indicate 10–90 percentiles, and horizontal lines denote the mean.
D     Correlation of erythrocyte count to the "contamination index" for the erythrocyte marker panel.
E     Correlation of platelet count to contamination index for the platelet marker panel.

coagulation factor F13A1 and antithrombin-III (SERPINC1) decreased by more than half. Interestingly, the strongest elevated proteins in serum were highly abundant platelet proteins: platelet basic protein (PPBP), platelet glycoprotein Ib alpha chain (GP1BA), thrombospondin 1 (THBS1), and platelet glycoprotein V (GP5) ($P < 10^{-10}$; twofold to fivefold increase). In total, 208 proteins increased and 91 decreased due to coagulation. The former set of proteins, which have higher levels in serum than in plasma, were also quantitatively enriched with high-abundant platelet proteins ($P < 10^{-5}$; median rank 699 of 3,150 proteins), indicating coagulation-induced activation of platelets.

To define a robust panel of quality markers for the extent of coagulation, we first selected the 30 most significantly altered proteins between serum and plasma. Although not among the top 30, we added the platelet factor 4 variant 1 (PF4v1; $P < 10^{-11}$, 2.2-fold up in serum), because it was an excellent indicator of

coagulation in our studies and has already been reported in the context of pre-analytical variation (Timms *et al*, 2007).

In contrast to the erythrocyte and platelet panels, proteins of the coagulation panel increase or decrease due to blood clotting and the fold changes vary strongly between them. Because fold changes are greatest for the decreasing proteins, we calculated the coagulation marker ratio only from them (sum of all plasma proteins divided by sum of plasma-elevated coagulation proteins). This ratio was very robust when comparing serum and plasma, clearly separating them with median ratios of 9 and 120 for these distinct sample types (Fig 3C). Of the coagulation marker panel, only F13A1, PPBP, and THBS1 were in common with the platelet panel and none with the erythrocyte panels (Fig 3D). The low overlap observed for the three quality marker panels should make them highly specific tools to elucidate the presence and origin of sample-related bias.

### Application of the quality marker panels to a biomarker study

The above-defined marker panels can assess sample-related issues at three levels: the quality of each sample in a clinical cohort, potential systematic bias in the entire study, and the likelihood that individual biomarker candidates belong to the contaminant proteomes.

We recently investigated changes in the plasma proteome upon weight loss (Geyer *et al*, 2016a,b). Briefly, caloric restriction in 52 individuals for 2 months was followed by weight maintenance for 1 year. Plasma Proteome Profiling of seven longitudinal samples revealed significant changes in the profile of apolipoproteins, a decrease in inflammatory proteins and markers correlating with insulin sensitivity. Given that protein abundance changes of < 20% were often highly significant, we expected that overall sample quality was high, making this study suitable for testing the practical applicability of the quality marker panels.

First, we assessed the quality of each sample separately by calculating the three contamination indices and plotting their distribution in the total of 318 measurements. For each index, we initially defined potentially contaminated samples as those with a value more than two standard deviations above the mean (red lines in Fig 4A). This flagged 12 samples, six with platelet contamination, one with increased erythrocyte levels, and five with signs of partial coagulation. Resolving the three quality marker panels to the levels of individual proteins resulted in almost perfectly parallel trajectories (Appendix Fig S4A–C). Accordingly, the correlations to the reference quality marker panels were substantial ($R > 0.77$). Overall, the variation of the contamination indices was highest for the platelets also visible by a contamination index difference (max/min ratio) of a factor 182 between the least and the most contaminated sample, followed by erythrocytes (max/min 23), and lowest for coagulation (max/min 5). The platelet proteins talin-1 (TLN1), myosin-9 (MYH9), and alpha-actinin-1 (ACTN1) had the largest variations, all with maximal changes > 5,000-fold. Catalase (CAT), carbonic anhydrase 1 and 2 (CA1, CA2) from the erythrocyte index varied maximally by more than 500-fold. The three fibrinogens in the coagulation panel changed by up to 20-fold, indicating that only partial coagulation events took place (Fig 4A).

Note that evaluating individual sample quality based on the standard deviation of all samples, as done here, has the benefit of being independent of the specific proteomic method used to measure protein amounts. However, this requires that most samples have

low levels of contamination, so that outliers of the statistical distribution are clearly apparent. If this is not the case, we propose using general, study-independent cutoff values to differentiate between samples of high and poor quality in such studies.

To assess potential systematic bias for groups of samples such as cases and controls or different time points, we applied a *t*-test based volcano plot. Most of the significantly upregulated proteins at time point 4 were members of the platelet panel (Fig 4B). With this information in hand, we contacted our collaboration partners, who tracked down the platelet contamination to a switch of the blood-taking equipment due to low supplies.

In practice, such sample issues will occasionally happen in a clinical study, and our quality marker panels would allow elimination of the affected samples. However, if contaminating proteins can reliably be distinguished from relevant biomarker candidates, the data could still be used. In our example, six of the eight significant outliers were from the platelet panel, and the other two proteins—GP1BA and NRP1—could still be of interest. To investigate this further, we inspected the global correlation map of all proteins, time points, and participants (Albrechtsen *et al*, 2018). In this hierarchical clustering analysis, proteins that are co-regulated have a high correlation to each other and appear in groups, visualized as red patches (Fig 4C). Here, the platelet cluster was the second largest one with 38 proteins ($R = 0.69$). All quantified platelet panel proteins were in this cluster, as was GP1BA, flagging them as likely contaminants (Fig 4C and inset). Interestingly, NRP1, a receptor involved in angiogenesis, did not group with the platelet proteins, suggesting a potential biological role. This is supported by the fact that NRP1 was significantly regulated over all time points compared to the baseline, in contrast to the platelet cluster proteins.

The other two quality marker panels are also readily apparent in the global correlation map. Ten members of the erythrocyte panel cluster tightly as do the three fibrinogen chains (Appendix Fig S5). However, in this study the fibrinogens group with proteins involved in low-grade inflammation, reduction of which was one of the main findings of our study (Appendix Fig S5). In contrast, the coagulation

marker PF4v1, which is also a highly abundant protein in platelets, clustered in the platelet group in this analysis, indicating that it varied as a result of sample preparation.

To make the above-described analysis readily available, we created an online platform at www.plasmaproteomeprofiling.org. It provides a toolbox for the interactive assessment of the quality of plasma proteomic data. Lists of protein abundances from MaxQuant search result tables or the template (Table EV4) can be uploaded by a simple drag and drop system. The system automatically generates the three contamination index values as shown in Fig 4A. If the user indicates cases and controls, the data set will be analyzed for systematic bias as visualized in a volcano plot (Fig 4B). The global correlation map is also displayed with the clusters of the quality marker panels (Fig 4C). The website is designed in the Dash data visualization framework, which allows further interactive analysis of the data (see Materials and Methods). Potential biomarker candidates in the volcano plot can be selected and displayed in the global correlation map to check whether the protein falls into or near one of the quality marker clusters.

### Revisiting results of published biomarker studies

Having examined one study in detail, we set out to survey the extent to which quality marker proteins are reported as biomarker candidates in the literature. To this end, we performed a comprehensive PubMed search requiring the terms 'proteomics', 'proteome', 'plasma OR serum', 'biomarker' and 'mass spectrometry' spanning the time frame from 2002 to April 2018. We excluded review papers, purely technological publications without biomarker candidates, animal studies, and publications without proteins as qualitative or quantitative variables. From the resulting 210 publications, we manually extracted the lists of the biomarker candidates that were reported as "significantly altered proteins" by the authors. Gene and protein names were mapped to the corresponding protein identifiers in our reference panels and analyzed for their frequencies.

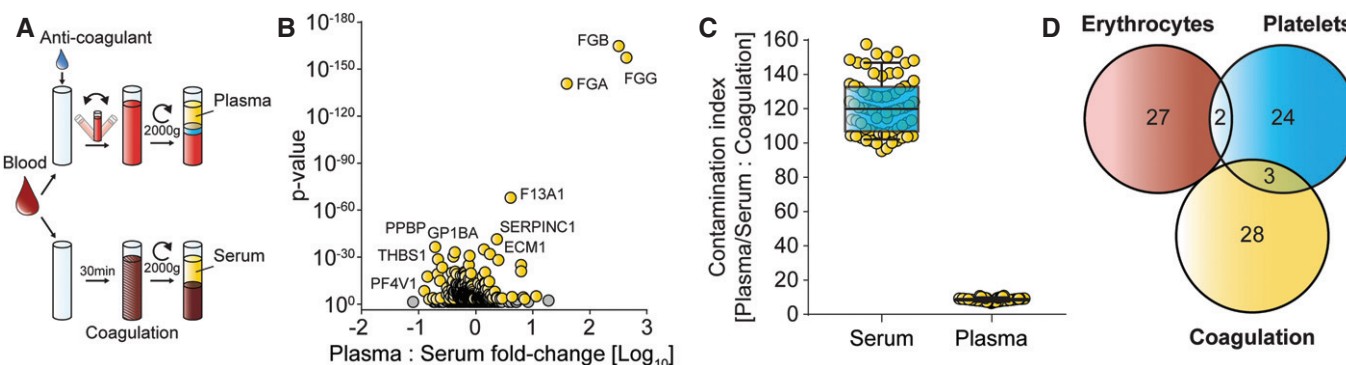

**Figure 3. Quality marker panel for blood coagulation.**

A Preparation of plasma and serum samples. EDTA was used as anticoagulation agent, and incubation and centrifugation values are indicated.

B Volcano plot comparing 72 plasma vs. 72 serum proteomes. Proteins highlighted in yellow were chosen according to their *P*-value as markers for coagulation. Only the plasma-enriched proteins (compared to serum) were used in the calculation of the coagulation contamination index.

C Ratio of the summed intensities of all plasma or serum proteins to the sum of the plasma-enriched panel proteins is plotted for all samples. Whiskers indicate the 10–90 percentile, and horizontal lines denote the mean.

D Overlap of the three quality marker panels.

Remarkably, 113 studies (54%) reported at least one potential quality marker as a biomarker candidate or as a statistically significant association (Fig 4D). As the total quality marker panel consists of 84 proteins and the median number of candidates per clinical study was seven, a certain overlap is not entirely unexpected. However, the candidates in question almost always were near the top of most abundant proteins of the quality marker panels, making it highly likely that they are indeed contaminants. Furthermore, while an individual protein could still be a genuine biomarker candidate, the fact that 22 studies (11%) reported two of them, and a further 23 studies (11%) three or more, again makes quality issues the likely explanation.

The majority of these studies reported proteins as potential biomarkers or as significant outliers of the coagulation panel, followed by the erythrocyte and platelet panels (Fig 4E). The most frequent one was clusterin (CLU; 27 times), followed by the fibrinogens (alpha, beta, and gamma; 22, 10, and 15 times), prothrombin (F2; 17 times), kininogen (KNG1; 15 times), antithrombin-III (SERPINC1; 13 times), and platelet basic protein (PPBP; 10 times). It is worth noting that proteins related to erythrocyte leakage may falsely be taken to indicate activation of oxidative pathways. For example, the hemoglobin subunits (e.g. HBA1, HBB, and HBD, listed 1, 6, and 1 time), carbonic anhydrases (CA1 and CA2, 6 and 6 times), fructose-bisphosphate aldolase (ALDOA, 5 times), peroxiredoxin 2 (PRDX2, 3 times), and superoxide dismutase (SOD1; 2 times) are annotated with keywords linked to oxidation. To illustrate this, a recent publication connected plasma proteome alterations in type 1 diabetes to oxidative stress. This may be a spurious link because the reported proteins were mostly members of the erythrocyte quality marker panel (Liu *et al*, 2018). Although platelet panel proteins are not prominent in the biomarker literature yet, we expect that they—along with lower abundant erythrocyte-specific proteins—will play an increasing role as technological progress enables higher plasma proteome coverage. We caution that platelet proteins already found in the biomarker literature such as PPBP, THBS1, and PF4 are often linked to coagulation events.

**Recommendations for future proteomic studies**

Based on our experience with the above-defined three quality marker panels (Table EV2) and analysis of thousands of plasma proteomes, we devised a general guideline for minimizing and detecting biases related to sample taking and processing (Table 1).

To further document the influence of common variables in the blood-taking process, we invited 10 healthy individuals and collected blood in 10 different blood sampling tubes. In this experiment, we systematically varied the type of plasma/serum, the blood specimen tubes (with or without gel), and the deposition of blood into the sampling tube (vacuum vs. pull system).

The most prominent differences were again between serum and plasma (Fig 3B; Appendix Fig S6). Apart from this, we found that contaminations with high-abundant erythrocyte-specific proteins appeared in several comparisons. Serum and EDTA plasma both had significantly higher levels than lithium heparin and citrate plasma (Appendix Fig S6A–F). Moreover, vacuum sampling can have an influence on erythrocyte-specific protein levels for some tubes. For instance, we found significantly increased levels of HBA1 and HBB in lithium heparin plasma tubes after vacuum sampling compared to a pull system, but not in the same comparison when using serum tubes (Appendix Fig S7A–D). Furthermore, erythrocyte-specific

proteins were significantly increased in lithium heparin pull tubes (more than twofold), which contain a gel plug compared to pull tubes without a gel plug (Appendix Fig S8A–D). In contrast, there were no differences between serum tubes with and without gel. These findings illustrate how even seemingly minor changes in blood-taking equipment can result in statistically significant differences of protein levels, which could confound biomarker studies. They also highlight the value of unbiased, system-wide investigation of the blood proteome and our quality marker panels.

We also found that the procedure of sampling the plasma from the tubes has a prominent effect on platelet contamination (Appendix Figs S9 and S10). Thus, we recommend not to collect the lowest layer of the plasma above the platelet bed after centrifugation. Furthermore, any delay from centrifugation to plasma harvest has the potential to induce platelet protein contamination. These factors mainly influence the platelet rather than the erythrocyte contamination index, indicating that proteins from the platelet proteome are the most likely cause of erroneous assignment of biomarker candidates.

## Discussion

Blood plasma remains the predominant biological matrix to assess health and disease in clinical settings. Around the world, every day hundreds of thousands of samples are analyzed to determine the levels of individual proteins. Likewise, blood plasma is directly or indirectly assessed in most clinical trials. Protein levels in plasma can readily be affected by cellular contamination or handling-related issues, and in clinical practice, this is partially addressed by simple tests such as those for hemoglobin contamination. However, these tests are not systematic or quantitative and they can only be used to exclude clearly contaminated samples.

Because of its high specificity and unbiased nature, MS-based proteomics is ideally suited to characterize the quality of blood plasma and it requires < 1 μl of material. So far, research on sample quality involving MS has mainly been restricted to the stability of internal standards in targeted assays and has rarely addressed overall sample quality (Schrohl *et al*, 2008; Hassis *et al*, 2015; Hoofnagle *et al*, 2016). Employing our Plasma Proteome Profiling pipeline to various clinical studies suggested that platelets, erythrocytes, and coagulation are by far the most important causes of plasma quality issues. We acquired very deep reference proteomes for these cell types and blood compartments, which we provide to the community to evaluate the possible origin of proteins emerging from biomarker studies. We defined three panels of about 30 proteins each that can serve as contamination indices (Table EV2). Using the example of a longitudinal Plasma Proteome Profiling study of weight loss and our online resource, we illustrated how the contamination indices can flag individual suspect samples and systematic biases. Furthermore, correlation analysis reveals whether potential biomarkers emerging from a given study are likely to be associated with quality-related proteome changes instead. Conversely, this procedure can "rescue" genuine biomarker candidates that are part of the quality marker proteomes. As an example, fibrinogens, a member of the coagulation quality marker panel, can also change during an inflammatory condition and might be correlated with classical inflammation markers such as CRP. In certain diseases, the entire set of proteins of a quality marker panel can be altered. For example, increased platelet

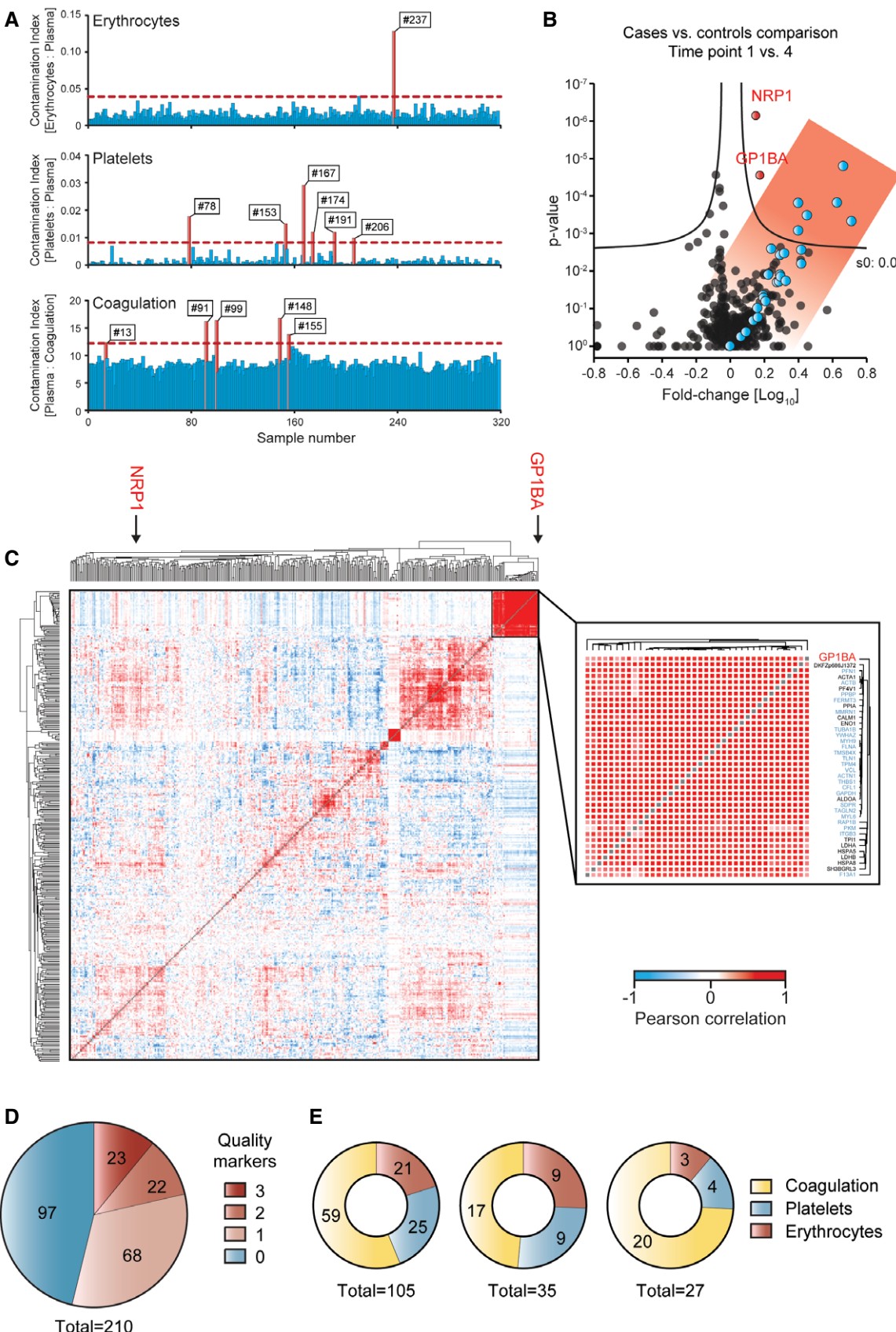

**Figure 4.**

◀

**Figure 4. Quality marker panels in a weight loss study and literature study.**

A Assessment of individual sample quality with respect to the three contamination indices using the online tool at www.plasmaproteomeprofiling.org. Samples with indices that are more than two standard deviations from the mean (horizontal red lines) are flagged as potentially contaminated (red bars and sample numbers).

B Volcano plot of the proteome comparison of time point 1 vs. 4. Proteins of the platelet panel are highlighted in blue and two additional significantly regulated proteins in red.

C Global correlation map on the left with an inset of the platelet cluster on the right. The two significant outliers of the volcano plot in (B) are marked in red. Platelet panel proteins are highlighted in blue in the inset. Red patches in the global correlation map indicate positive and blue patches negative correlations.

D Literature analysis of 210 publications using MS-based plasma proteomics to identify new biomarkers. The number of quality markers reported as biomarker candidates in these studies is indicated.

E Distribution of the reported quality markers according to the three types of likely contaminations. The distribution is shown across studies that report one, two, or three proteins of the same quality marker panel.

levels—thrombocythemia—can have a variety of causes ranging from chronic inflammation to myeloproliferative diseases. Likewise, increased concentration of erythrocyte-specific proteins can be caused by hemolytic diseases such as in autoimmunity. While these cases are not the usual reasons why a quality marker panel is altered, they need to be considered when judging the analytical validity of a plasma measurement.

The clinical potential of the plasma proteome has long been realized and is also emphasized by the fact that more than 50

**Table 1. Practical considerations to minimize systematic bias.**

| General instructions |
| --- |
| Avoid pooling of samples |
| Use plasma or serum exclusively, not a combination |
| **Sample collection** |
| Standardize blood collection and pre-analytical procedures (preferably same person collecting blood, centrifuge, sampling container, storage temperature, and time) |
| Centrifuge blood to generate plasma immediately |
| Centrifuge according to manufacturer's instruction |
| Harvest plasma immediately after centrifugation |
| Harvest the plasma starting from the top of the container and pool it before aliquotting |
| Discard the last 500 μl of plasma to avoid contamination with platelets or use a second centrifugation step to generate platelet-poor plasma |
| Freeze samples immediately after harvesting |
| **Principal assessment of study sample quality** |
| When working with a new batch of samples from collaborators: run at least 10 test samples of each study group by mass spectrometry |
| Use quality marker panels to check for any indication of contamination |
| **Main study** |
| Continuously assess quality during the project to detect and avoid systematic bias (pre-analytics, mass spectrometric analyses) |
| Overall quality: report the number of contaminated samples |
| Systematic bias: report potential systematic bias |
| Check whether biomarker candidates are contained in the quality marker panels |
| Identification of several quality markers as biomarker candidates may be indicative of a study vector |
| If a quality marker is among the biomarker candidates, thorough validation is required |

FDA-approved biomarkers can be quantified even in relatively shallow proteomic measurements of plasma (Geyer *et al*, 2016a). If there are as many new biomarkers among the less abundant proteins, there should be a diagnostic treasure trove still to be discovered (Geyer *et al*, 2017). Millions of plasma samples are stored in biobanks worldwide, representing an immense untapped resource that could be analyzed by MS-based proteomics or large-scale affinity-based methods. Despite initial enthusiasm and community efforts such as the Human Proteome Organization's plasma proteomic initiative (Omenn *et al*, 2005; Schwenk *et al*, 2017), few if any new protein biomarkers have entered the clinic in recent decades. This is probably at least partially due to techno-logical limitations to characterize the vast dynamic range of the plasma proteome, which in turn has led to underpowered study designs (Geyer *et al*, 2017). While many of these challenges are already being addressed, we suspect that problems with sample qual-ity represent another important reason for the paucity of new biomarkers and, even more seriously, for incorrect biomarkers being used. Examining our own data as well as the scientific literature, we here show that sample quality issues indeed have an impact on reported results. Nearly half of the reviewed studies reported at least one potential biomarker that is in our quality marker panels, and many had two or more, making sample contamination very likely. While coagulation-related issues are currently most prominent, increasing depth of plasma proteome coverage may replace platelet contamination as the most important source of error in the future. A corollary of the very large abundance variation of proteins introduced by quality issues is that it should further discourage pooling of samples. While this increases throughput, even a single contaminated sample can readily skew an entire batch.

Systematic bias introduced by imperfect sample handling or processing may lead to reporting incorrect biomarkers. Conversely, randomly distributed samples with poor quality will diminish over-all statistical quality and may obscure true biomarker candidates.

The sources of quality issues are different kinds of variations in the pre-analytical processes, and we found platelet contamination during plasma harvesting to be one of the main culprits. Among the few previous studies, Hassis *et al* (2015) investigated different sample handling errors and concluded that only extreme conditions, such as delay in sample storage for 4 days, substantially changed the plasma proteome. However, proceeding with such extreme cases is rare, and quality issues are much more likely to originate from recontamination with whole blood after centrifugation during the plasma harvest or post-centrifugation times and resuspension of platelets, for instance. The comparison of 10 different blood sampling tubes showed that even seemingly minor differences in

the sample handling devices like a pull vs. a vacuum deposition system can have a statistically significant effect on the measured proteome. Therefore, we want to stress the importance of strictly following standard operating procedures. We here provide general considerations for minimizing sample-related issues, ranging from immediate harvest of the plasma after centrifugation to discarding the lowest layer of plasma to avoid recontamination with platelets (Table 1). These recommendations update and extend general good laboratory practices as well as HUPO guidelines (Omenn *et al*, 2005; Rai *et al*, 2005). We also advocate that plasma samples are quality-checked by MS-based proteomics, at least for a representative subset. This is especially important for clinical studies but also for targeted single-analyte measurements, which by their nature are blind to the overall composition of the sample. Although it would be possible to determine contamination indices by multiplexed affinity-based methods, we recommend MS for this purpose because of its very high specificity and its unbiased nature. Furthermore, the proteomic depth needed to assess the quality is easily achievable even in rapid and economical measurements.

The concepts and methods put forward in this study could readily be adapted to other body fluids such as urine, saliva, or cerebrospinal fluid. This would require developing the appropriate contamination indices. Furthermore, the three quality marker categories are the largest but not the only ones. For instance, we imagine that similar experiments can be performed to gauge the effect of storage duration and temperature on the plasma proteome as it influences MS-based proteomics.

In conclusion, sample-related quality issues are clearly a concern for biomarker studies. However, we show here that they can be addressed rigorously and comprehensively by MS-based proteomics. As this technology continues to improve in throughput, depth, and robustness, we envision that it will be employed in routine clinical practice. Biomarker panels instead of single markers will be measured by MS-based proteomics as this takes advantage of its inherently multiplexed nature and allows the characterization of clinical conditions more comprehensively. These biomarker panels could routinely be extended with quality marker panels as introduced here, helping to establish biomarker-guided decisions in a wide variety of clinically important areas.

## Materials and Methods

### Samples for defining the three quality marker panels

All participants gave written informed consent for their participation in the Munich Study on Biomarker Reference Values (MyRef), which is registered under the local ethic number 11-16. All experiments conformed to the principles set out in the WMA Declaration of Helsinki and the Department of Health and Human Services Belmont Report.

To establish the quality marker panels, whole blood was harvested by venipuncture of 10 females and 10 males into commercial EDTA-containing sampling containers. The blood was centrifuged at 200 *g* for 10 min, and both the pellet and the supernatant were kept for further processing steps. The bottom layer of 500 µl plasma was discarded to avoid contamination of the platelet-rich plasma fraction with erythrocytes. The pellet was centrifuged at 2,000 *g* for 15 min, and the top layer containing plasma, the buffy coat, and 1 ml of erythrocytes were discarded. After adding 4 ml PBS containing 1.6 mg/ml EDTA, the suspension was centrifuged at 2,000 *g* for 15 min and the supernatant was discarded together with 500 µl of the top layer of the erythrocytes. This step was repeated, and the pure erythrocyte fraction was harvested. We centrifuged the supernatant from the first centrifugation step containing plasma and platelets a second time at 200 *g* for 10 min and harvested the supernatant, which constitutes the platelet-rich plasma. This step was repeated, and we collected the supernatant and the platelet after centrifugation at 2,000 *g* for 15 min. The supernatant was centrifuged a second time at 2,000 *g* for 15 min to harvest platelet-free plasma by sampling only top layer of the supernatant, but discarding the bottom layer of 500 µl. The platelets were washed twice by adding 4 ml PBS containing 1.6 mg/ml EDTA and centrifugation at 2,000 *g* for 15 min. The supernatant was discarded, and the pure platelet fraction was harvested.

For the serum and plasma comparison, blood samples from two females and two males were split into 18 samples each and serum and plasma were harvested after centrifugation at 2,000 *g* for 15 min.

To investigate the effects of different blood sampling devices on the blood plasma proteome, we invited 10 healthy individuals (five female and five males) and collected blood in the 10 different blood sampling devices (Table EV5). After collecting whole blood, it was incubated at room temperature for 30 min to allow coagulation in the serum tubes. The plasma tubes were also stored at room temperature for the same time, and the different tubes were centrifuged together. Afterward, 0.5 ml of plasma or serum was sampled from the top of the tubes.

To evaluate the platelet contamination in different layers of plasma after centrifugation, blood was collected in two different 9-ml S-Monovette EDTA-containing sampling containers (Sarstedt). The blood of one container was transferred to a 15-ml centrifugation tube without separation gel. Both containers were centrifuged at 2,000 *g* for 15 min. Plasma was harvested in nine volume fractions starting from the top layer in 500 µl steps to the top of the buffy coat. The buffy coat itself was not touched, and a small amount of plasma (~200 µl) remained on top.

### High-abundant protein depletion for building a matching library

We created a matching library and applied a consecutive depletion strategy, in which the top 6 and top 14 most abundant plasma proteins were depleted by using a combination of two immunodepletion kits, as described in ref. Geyer *et al* (2016a). Briefly, the Agilent Multiple Affinity Removal Spin Cartridge was used for the depletion of the top six highest abundant proteins (albumin, IgG, IgA, antitrypsin, transferrin, and haptoglobin), followed by Seppro Human 14 Sigma immunodepletion for the 14 highest abundant proteins (albumin, IgG, IgA, IgM, IgD, transferrin, fibrinogen, α2-macroglobulin, α1-antitrypsin, haptoglobin, α1-acid glycoprotein, ceruloplasmin, apolipoprotein A-I, apolipoprotein A-II, apolipoprotein B, complement C1q, complement C3, complement C4, plasminogen, and prealbumin). Following depletion, we fractionated our samples using the high pH

reversed-phase "Spider fractionator" into 24 fractions as described previously (Kulak *et al*, 2017).

The paper explained

**Problem**

New biomarkers are urgently needed in many health and disease contexts and mass spectrometry-based proteomics is a potentially powerful and promising technology for their discovery, as it can analyze the plasma proteome in a quantitative and specific manner. However, a systematic analysis of pre-analytical variations might obscure the discovery of novel biomarkers and has not been performed so far.

**Impact**

Quality issues due to erythrocyte lysis, platelet contamination, and partial coagulation might affect up to 50% of all biomarker studies as we showed by a literature survey of more than 200 published manuscripts. Our quality marker panels will prevent costly miss-assignment of potential biomarker candidates and support the discovery of promising biomarkers.

## Sample preparation: protein digestion and in-StageTip purification

Sample preparation was carried out according to our Plasma Proteome Profiling pipeline as described in Geyer *et al* (2016a,b) with an automated setup on an Agilent Bravo Liquid Handling Platform. In brief, plasma samples were diluted 1:10 with $_{dd}H_2O$ and 10 μl of the sample was mixed with 10 μl PreOmics lysis buffer (P.O. 00001, PreOmics GmbH) for reduction of disulfide bridges, cysteine alkylation, and protein denaturation at 95°C for 10 min (Kulak *et al*, 2014). Trypsin and LysC were added to the mixture after a 5-min cooling step at room temperature, at a ratio of 1:100 micrograms of enzyme to micrograms of protein. Digestion was performed at 37°C for 1 h. An amount of 20 μg of peptides was loaded on two 14-gauge StageTip plugs, followed by consecutive purification steps according to the PreOmics iST protocol (www.preomics.com). The StageTips were centrifuged using an in-house 3D-printed StageTip centrifugal device at 1,500 *g*. The collected material was completely dried using a SpeedVac centrifuge at 60°C (Eppendorf, Concentrator plus). Peptides were suspended in buffer A* [2% acetonitrile (v/v), 0.1% formic acid (v/v)] and sonicated (Branson Ultrasonics, Ultrasonic Cleaner Model 2510). Pools for each of the five sample types (whole blood, erythrocytes, platelets, plasma, and platelet-free plasma) were generated from the 20 individuals and prepared according to the procedure above. The peptides were fractionated using the high pH reversed-phase "Spider fractionator" into 24 fractions as described previously to generate deep proteomes (Kulak *et al*, 2017).

## Ultra-high-pressure liquid chromatography and mass spectrometry

Samples were measured using LC-MS instrumentation consisting of an EASY-nLC 1000 or 1200 ultra-high-pressure system (Thermo Fisher Scientific), which was coupled to a Q Exactive HF Orbitrap (Thermo Fisher Scientific) using a nano-electrospray ion source (Thermo Fisher Scientific). Purified peptides were separated on 40-cm HPLC columns [ID: 75 μm; in-house packed into the tip with ReproSil-Pur C18-AQ 1.9 μm resin (Dr. Maisch GmbH)]. For each LC-MS/MS analysis, about 0.5 μg peptides were used for 45-min runs and for each fraction of the deep plasma data set.

Peptides were loaded in buffer A [0.1% formic acid and 5% DMSO (v/v)] and eluted with a linear 35-min gradient of 3–30% of buffer B [0.1% formic acid, 5% DMSO, and 80% (v/v) acetonitrile], followed stepwise by a 7-min increase to 75% of buffer B and a 1-min increase to 98% of buffer B, followed by a 2-min wash of 98% buffer B at a flow rate of 450 nl/min. Column temperature was kept at 60°C by an in-house-developed oven containing a Peltier element, and parameters were monitored in real time by the SprayQC software (Scheltema & Mann, 2012). MS data were acquired with a Top15 data-dependent MS/MS scan method for the construction of the library and BoxCar scans (Meier *et al*, 2018) for the study samples. Target values for the full-scan MS spectra were $3 \times 10^6$ charges in the 300–1,650 m/z range with a maximum injection time of 55 ms and a resolution of 60,000 at m/z 200. Fragmentation of precursor ions was performed by higher-energy C-trap dissociation (HCD) with a normalized collision energy of 27 eV. MS/MS scans were performed at a resolution of 30,000 at m/z 200 with an ion target value of $1 \times 10^5$ and a maximum injection time of 120 ms. Dynamic exclusion was set to 30 s to avoid repeated sequencing of identical peptides.

## Data analysis

MS raw files were analyzed by MaxQuant software, version 1.5.6.8, (Cox & Mann, 2008), and peptide lists were searched against the human UniProt FASTA database. A contaminant database generated by the Andromeda search engine (Cox *et al*, 2011) was configured with cysteine carbamidomethylation as a fixed modification and N-terminal acetylation and methionine oxidation as variable modifications. We set the false discovery rate (FDR) to 0.01 for protein and peptide levels with a minimum length of 7 amino acids for peptides, and the FDR was determined by searching a reverse database. Enzyme specificity was set as C-terminal to arginine and lysine as expected using trypsin and LysC as proteases. A maximum of two missed cleavages were allowed. Peptide identification was performed with an initial precursor mass deviation up to 7 ppm and a fragment mass deviation of 20 ppm. The "match between run algorithm" in the MaxQuant quantification (Nagaraj *et al*, 2012) was enabled after constructing a matching library consistent of depleted and all the undepleted plasma samples. All proteins and peptides matching to the reversed database were filtered out. Label-free protein quantitation (LFQ) was performed with a minimum ratio count of 2 (Cox *et al*, 2014).

## Bioinformatic analysis

All bioinformatic analyses were performed with the Perseus software of the MaxQuant computational platform (Cox & Mann, 2008;

Tyanova *et al*, 2016). For the global correlation analysis, proteins were filtered for at least 50% valid values in the weight loss study and the hierarchical clustering was performed using Euclidean distance. The weight loss study contained in total 28 proteins of the platelet panel, but after sorting for 50% valid values only 24 were left and all of them clustered in the platelet panel.

### Online platform for automated analysis of clinical studies

Our online portal is equipped with a user-friendly graphical interface that supports the most common web browsers, such as Google Chrome, Firefox, and Internet Explorer. For the front-end development, a Dash framework was used (version 0.27.0), which consists of a Flask server (1.0.2) that communicates with front-end React.js components using JSON, or JavaScript Object Notation, packets (a minimal, readable format for structuring data) over HTTP, or Hypertext Transfer Protocol, requests that work as request–response protocols between a client and server. Taking advantage of the full power of Cascading Style Sheets (CSS), every graphical element was customized: the sizing, the positioning, the colors, and the fonts.

The platform takes the results of the MS data processed by the MaxQuant software (Cox & Mann, 2008) from the proteinGroups table (to be extended to other formats). During the data uploading, the input file is verified through a combination of preliminary tests. We built a complex data structure using general Python libraries, such as NumPy, Pandas, and SciPy. Using three panels of markers for platelet contamination, erythrocyte contamination, and coagulation events in plasma samples, respectively, we identify samples affected by quality issues. Samples having at least 50% "valid values" (i.e. those with quantification results) are preprocessed by cleaning the data and prepare them for the subsequent visualization step.

## Data availability

The MS-based proteomic data have been deposited to the ProteomeXchange Consortium via the PRIDE partner repository and are available via ProteomeXchange with identifier PXD011749 (https://www.ebi.ac.uk/pride/archive/projects/PXD011749).

**Expanded View** for this article is available online.

### Acknowledgements
We thank all members of the Proteomics and Signal Transduction Group and the Clinical Proteomics Group for help and discussions and in particular Igor Paron, Christian Deiml, Alexander Strasser, and Gaby Sowa for technical assistance; Mario Oroshi for help with the online resource; Nicolai J. Wewer Albrechtsen, Nils A. Kulak, Niels Skotte, and Martin Steger for discussion; and Jürgen Cox for bioinformatic tools. The work carried out in this project was partially supported by the Max Planck Society for the Advancement of Science, the European Union's Horizon 2020 research and innovation program with the MSmed project (no. 686547), and grants from the Novo Nordisk Foundation (NNF15CC0001; NNF15OC0016692) and the BMBF grant German Biobank Alliance (BMBF 01EY1711C).

### Author contributions
PEG designed, performed, and interpreted the MS-based proteomic analysis of patient plasma; wrote the paper; and generated the figures. PVT wrote the manuscript and performed together with LN, SD, JBM, AK, MLB, and JB experiments and generated article text. DT and LMH designed experiments, drafted practical considerations for sample preparation, and worked on the article text. EV designed and established the interactive online resource. MM designed and interpreted the MS-based proteomic analysis of plasma, supervised and guided the project, and wrote the manuscript.

### Conflict of interest
The authors declare that they have no conflict of interest.

### For more information
(i)   https://www.biochem.mpg.de/en/rd/mann
(ii)  https://www.cpr.ku.dk/research/proteomics/mann-group/
(iii) http://www.plasmaproteomeprofiling.org/

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
