## [Review Process File · EMBO Molecular Medicine]

Plasma proteome profiling to detect and avoid sample-related biases in biomarker studies

Philipp E. Geyer, Eugenia Voytik, Peter V. Treit, Sophia Doll, Alisa Kleinhempel, Lili Niu, Johannes B. Müller, Marie-Luise Buchholtz, Jakob Bader, Daniel Teupser, Lesca M. Holdt and Matthias Mann.

Review timeline:

Submission date:	4 th February 2019
Editorial Decision:	8 th March 2019
Revision received:	28 th June 2019
Editorial Decision:	19 th August 2019
Revision received:	26 th August 2019
Accept:	3 rd September 2019

Editor: Jingyi Hou

Transaction Report:

1st Editorial Decision

8th March 2019

Thank you for the submission of your manuscript to EMBO Molecular Medicine. We have now heard back from the three referees whom we asked to evaluate your manuscript. As you will see below, the reviewers find the manuscript to be of interest, providing technical quality and clinical value. They also have suggestions and recommendations to further improve the data presentation, conclusiveness and clarity.

Without repeating all the points raised in the reviews below, we would strongly encourage you to investigate heparine and citrated plasma matrices as recommended by reviewer #1 to improve the clinical significance of the study. Please feel free to contact me in case you would like to discuss in further detail any of the issues raised by the reviewers.

REFeree REPORTS

Referee #1 (Comments on Novelty/Model System for Author):

The authors studied the impact of the preanalytical phase on the quality of clinical samples meant to be used for biomarker discovery with in depth proteomics. The authors documented cellular contamination of EDTA-plasma from lysed erythrocytes and/or platelets, and also found significant impact of coagulation. The consequences of these findings are very relevant as contamination of plasma proteins with ery-derived or platelet-derived proteins caused by pre-analytical processing leads to false conclusions about so-called promising candidate biomarkers and finally to irreproducible results and research waste. The problem is especially hampering proteomics discovery studies, but also targeted proteomics.

The authors have currently investigated serum and EDTA-plasma matrices. In routine medical labs, heparineplasma and citrateplasma are the matrices of choice for measuring conventional proteins

respectively coagulation factors. In addition, the matrix is always a part of the entire measurement system and has to be validated as an essential part of the total test application. As many biobanks also yield heparine- and citratedplasma, these matrices should also be investigated. Due to the granularity of MS both time and storage conditions before freezing affect the degree of contamination. In addition, the specimen tube (with gel, with preservatives ...) the vacuum sampling (yes/no), straight needle (yes/no), the transport systems used are also essential. All preanalytical variables should be investigated in a very strict and standardized way, but taking into account the routine practices of medical laboratories.

Spiking experiment: the question is whether this model is a suitable mimic for evaluating the effect of platelet contamination. After all, platelets are extremely sensitive to their environment and manipulation. This limitation should be mentioned.

Typo error? in the M&M it is stated at the top of p.16 that the blood is centrifuged at 200g for 10 min. Is this 200g or 2000g?

Referee #1 (Remarks for Author):

Standardization of the preanalytical phase is extremely relevant for discovery proteomics in order to assure that the selected candidate biomarkers are promising and not an artefact due to contamination of plasma (or serum). Setting up an EQA-website with the proposed three quality panels is helpful in order to investigate whether processed biobank samples from research institutes do qualify for the intended research and research questions. Establishing much more comprehensive recommendations for preanalytics seems to be needed. Also, more research is needed to understand the effect of all preanalytical variables (including transport, different matrices, temperature, freezing and thawing, ...).

Referee #2 (Comments on Novelty/Model System for Author):

1. The technical quality is high the experiments were designed and correctly performed to make the suggested conclusions.
2. The subject is inherently not novel since numerous publications have already suggested quality controls that can be used to assess plasma sample quality. However, an automated interface where proteomics data can be rapidly screened for sample quality is new as is the ability to designate the source of contamination originating from erythrocytes, platelets, the coagulation cascade or a combination of the three.
3. It is difficult to assess how popular this research tool will be. The medical impact ranges from medium to high.
4. The model system is identical to the system under study (human plasma) and therefore adequate.

Referee #2 (Remarks for Author):

Reviewer summary of the research article:

The manuscript entitled "Plasma proteome profiling to detect and avoid sample-related biases in biomarker studies" By Geyer et al., develops 3 quality indices that can be used to assess pre-analytical sample quality when using plasma for protein biomarker discovery. The three quality indices can be used to identify the source of protein contamination in plasma from erythrocytes, platelets and/or coagulation specific proteins. The quality indices were developed by first

performing label-free protein quantification on erythrocytes, platelets, platelet rich plasma, plasma, and serum, all of which were obtained from blood samples donated by 10 males and 10 females. To obtain quality indices that would identify contamination of plasma by proteins from erythrocytes or platelets, proteins which displayed a 10 fold higher intensity in erythrocytes or platelets (compared to plasma) and that had a CV < 30% across all individual samples were selected for quality marker panels. This approach identified 27 unique proteins to erythrocytes and 24 unique proteins to platelets. Subsequently, a set of dilution series was carried ranging from 1:1 erythrocyte/platelet:plasma down to 7 orders-of-magnitude dilution of cells. As expected, this resulted in a decrease of cellular proteins intensity and summing the protein intensities from each panel and dividing each by the total protein intensities in plasma provided two "quality indices" that were linear with respect to cell numbers measured by flow cytometry.

Similarly, to obtain a panel of proteins that could be used to assess systemic bias due to partial coagulation events, 72 plasma and 72 serum samples from 4 individuals were compared via label-free quantitative proteomics. The 30 most significantly altered proteins between serum and plasma were selected, including platelet factor 4 variant 1 because it is an indicator of coagulation. Similar to the aforementioned quality indices, the total protein intensity in plasma was divided by the sum of protein intensities of the 31 protein panel to obtain a coagulation specific quality index which could be used to clearly differentiate plasma from serum. Using these quality indices, the authors then assessed the quality of plasma samples used to perform a previous study in 2016 (Mol Syst Biol 12: 901). Across all 318 plasma samples, the ratio for each quality index was calculated and a ratio higher than 2 standard deviations was used as a cut-off to identify plasma samples that had not been prepared appropriately. Using the protein panels revealed that a number of the plasma samples had been contaminated. The authors also performed an extensive literature search of studies which used plasma for protein biomarker discovery and re-examined the datasets from 210 studies. This approach revealed 113 studies have reported at least one potential quality marker as a biomarker candidate. Based on these results, the authors provide recommendations for sample preparation and make the analysis readily available by creating an online platform at www.plasmaproteomeprofiling.org. Lists of protein intensities can be uploaded which automatically generates the three quality index values, systemic bias and a global correlation map.

Reviewer recommendation:

The developed website which streamlines assessment of pre-analytical plasma sample quality will aid researchers who use plasma for protein biomarker discovery. The automated interface is new, as is being able to define whether the contamination is from erythrocytes, platelets, or incomplete coagulation, or a combination of the three. While other studies make suggestions that quality controls should be included, the present study addresses the subject in greater depth. The recommendation is to publish with the following minor modifications:

- Page 3 on the last line change "low abundant" to "low concentration" or "low-abundance"
- Page 4 the last paragraph could be improved for clarity. The second line "several hundred plasma samples" I assume this is referring to the 2016b study by Geyer? It is not clear which plasma samples are being discussed as they are not prepared in the present study. This should be explicitly stated. The same is true for the "the study plasma samples" in the same paragraph on line four.
- The figure legend in supplemental Fig. S3 is not properly labeled; panels A and B are not designated in the legend.
- Page 5 "Spiked-in experiments" in the heading, is this valid terminology? Is it not more accurate to say serial dilution experiments validate the erythrocyte and platelet quality marker panels?
- Fig. 2 in the legend could you please state how the Z-score was calculated, it is not obvious.
- Page 7 in the second paragraph "Interestingly, the strongest elevated proteins in serum were all connected to platelets:" It is not clear what is meant by "connected to platelets" are they proteins unique to platelets?
- Page 7 second paragraph "In total, 208 proteins increased and 91 decreased due to coagulation." In plasma or in serum? this should be explicitly communicated here.
- Figure 4A, the y axes is labelled as contamination index but the figure legend and throughout the text these values are described as quality indexes. What is the difference?
- I attempted to use the online platform at www.plasmaproteomeprofiling.org. However, the instructions were not clear on how the text file should be formatted, and I did not succeed in using the platform. I think an example of the formatting or even a template would be helpful.
- Page 16 this sentence should be the starting sentence in the first paragraph, not the second as currently structured. "All participants gave written informed consent for their participation in the Munich Study on Biomarker Reference Values (MyRef), which is registered under the local ethic number 11-16."

-Page 16 please state how much blood was collected from each individual.

Referee #3 (Comments on Novelty/Model System for Author):

No model systems used.

Referee #3 (Remarks for Author):

EMM-2019-10427

Plasma proteome profiling to detect and avoid sample-related biases in biomarker studies
Philipp Geyer et al.

Geyer et al. present a new resource to identify contaminant proteins in human plasma proteomics studies. By acquiring deep reference proteomes, Geyer et al construct quality marker panels for contamination of erythrocytes, contamination of platelets, and coagulation. The study is of clear interest to the medical and biomarker communities. The manuscript is well written and generally easy to follow, and the authors have provided an online resource to assist the community with screening their plasma proteomes for contaminant proteins. I believe there are only a few issues which should be addressed (as described below).

Major issues:

- Why are there so many proteins with platelet / plasma and erythrocyte / plasma ratios of 10^7 in Fig. 1D and 1E? Please explain.
- It is unclear how we should interpret data from the fibrinogens (FGA, FGB, FGG). These proteins are included in the coagulation panel (Fig. 3B), and their disappearance should indicate coagulation. Yet, the authors appear to conclude that the fibrinogens might be biologically relevant in their own biomarker study data because "the fibrinogens group with proteins involved in low-grade inflammation" in the global correlation map (page 9). Does this mean that the fibrinogens are biomarkers of inflammation in this study? If so, how can the authors reconcile this conclusion in their own data with the claim that in many other biomarker studies that fibrinogens are frequently misidentified as biomarkers (page 12)?
- The authors claim on page 9 that "The other two quality marker panels are also readily apparent in the global correlation map", but I do not see the coagulation panel in either Fig. 4C or Fig. S5. The authors should highlight the coagulation panel as well in one of these figures.
- I could not find Supplementary Table S1.
- In general, the supplementary tables are not readable because they are spread out over many pages. Several pages in the manuscript pdf are blank because of this problem. The authors should provide this supplementary table in a more readable format, perhaps as an Excel file.

Minor issues:

- page 4, glyceraldehyde 3-phosphate dehydrogenase has gene symbol "GAPDH" not "GPDH". It is correct in Supp. Table S2.
- page 2, 2nd paragraph, should be "Within its limit of detection" not "Within its' limit of detection" (no apostrophe).
- page 4, "their quantities were not correlate with each other" should be "their quantities were not correlated with each other".
- page 11, "to survey the extent to which probably quality marker proteins are reported...", is the word "probably" necessary?

Point-by-point answers of reviewer's comments

We thank the reviewers for the in-depth and insightful comments on our manuscript “**Plasma proteome profiling to detect and avoid sample-related biases in biomarker studies**”.

We appreciate the positive evaluation of our efforts for improving the success rate of biomarker research by plasma proteomics, which we hope to achieve this with quality marker panels to evaluate individual sample quality, complete studies and biomarker candidates themselves. We agree with the recommendations from the reviewers and have answered them below and incorporated them in the revised manuscript. In particular, Reviewer #1 encouraged us to add new experiments that will be of general interest for the community such as the comparison of citrate, heparin and EDTA plasma, comparison of vacuum systems and the specimen tubes themselves. For this purpose, we collected a new set consisting of 100 samples from ten individuals to investigate the influence of all the above-mentioned factors on the plasma proteome. These results clearly highlighted how important it is to follow standard operating procedures (SOP) as even small changes in the sampling devices can have significant influences on the plasma proteome, which we show with the quality marker panels.

In total, we added three supplemental figures and two supplemental tables that show the influence of blood sample processing and plasma/serum collection. The new supplemental table S4 is a template to submit the data to plasmaproteomeprofiling.org, which will make it easy to submit data from search engines other than MaxQuant to the online platform.

Referee #1 :

The authors studied the impact of the preanalytical phase on the quality of clinical samples meant to be used for biomarker discovery with in depth proteomics. The authors documented cellular contamination of EDTA-plasma from lysed erythrocytes and/or platelets, and also found significant impact of coagulation. The consequences of these findings are very relevant as contamination of plasma proteins with ery-derived or platelet-derived proteins caused by pre-analytical processing leads to false conclusions about so-called promising candidate biomarkers and finally to irreproducible results and research waste. The problem is especially hampering proteomics discovery studies, but also targeted proteomics.

Author’s response: We thank the reviewer for the positive and constructive comments and we appreciate that he or she values our work with respect to biomarker research and systematic bias in clinical studies.

The authors have currently investigated serum and EDTA-plasma matrices. In routine medical labs,

heparineplasma and citrateplasma are the matrices of choice for measuring conventional proteins respectively coagulation factors. In addition, the matrix is always a part of the entire measurement system and has to be validated as an essential part of the total test application. As many biobanks also yield heparine- and citratedplasma, these matrices should also be investigated. Due to the granularity of MS both time and storage conditions before freezing affect the degree of contamination. In addition, the specimen tube (with gel, with preservatives ...) the vacuum sampling (yes/no), straight needle (yes/no), the transport systems used are also essential. All preanalytical variables should be investigated in a very strict and standardized way, but taking into account the routine practices of medical laboratories.

Author's response: We agree with the reviewer that additional experiments to elucidate the reasons for the contaminations would be valuable for the biomarker discovery field. We now include several new experiments. In the new Supplemental Fig. S6A-F, we cross-compared serum, lithium heparin, EDTA and sodium citrate plasma. The difference between the plasma samples and serum was as prominent as that between plasma and serum shown in figure 3B. Moreover, almost all the top outlier proteins like the fibrinogens FGA, FGB, FGG, SERPINC1 and F13A1 were decreased in serum compared to plasma whereas PPBP and THBS1 again increased. Additionally, we found effects on the levels of erythrocyte specific proteins, which we describe in a new paragraph:

To further document the influence of common variables in the blood taking process, we invited ten healthy individuals and collected blood in ten different blood-sampling tubes. In this experiment, we systematically varied the type of plasma/serum, the blood specimen tubes (with or without gel) and the deposition of blood into the sampling tube (vacuum vs. pull system).

The most prominent differences were again between serum and plasma (Fig. 3B; Supplemental Fig. S6). Apart from this, we found that contaminations with high abundant erythrocyte specific proteins appeared in several comparisons. Serum and EDTA plasma both had significantly higher levels than lithium heparin and citrate plasma (Supplemental Fig. S6A-F). Moreover, vacuum sampling can have an influence on erythrocyte specific protein levels for some tubes. For instance, we found significantly increased levels of HBA1 and HBB in lithium heparin plasma tubes after vacuum sampling compared to a pull system, but not in the same comparison when using serum tubes (Supplemental Fig. S7A-D). Furthermore, erythrocyte specific proteins were significantly increased in lithium heparin pull-tubes (more than two-fold), which contain a gel plug with respect to pull-tubes, which do not (Supplemental Fig. S8A-D). In contrast, there were no differences between serum tubes with and without gel. These findings illustrate how even seemingly minor changes in blood taking equipment can result in statistically significant differences of protein levels, which could confound biomarker studies. They also highlight the value of unbiased, systems wide investigation of the blood proteome and of our quality marker panels.

We also found that the procedure of sampling the plasma from the tubes has a prominent effect on platelet contamination (Supplemental Fig. S9, Fig. S10). Thus, we recommend not to collect the lowest layer of the plasma above the platelet bed after centrifugation. Furthermore, any delay from centrifugation to plasma harvest has the potential to induce platelet protein contamination. These factors mainly influence the platelet rather than the erythrocyte contamination index, indicating that proteins from the platelet proteome are the most likely cause of erroneous assignment of biomarker candidates.

Spiking experiment: the question is whether this model is a suitable mimic for evaluating the effect of platelet contamination. After all, platelets are extremely sensitive to their environment and manipulation. This limitation should be mentioned.

Author's response: We think that the spike-in experiments are very valuable as they are key experiments to document the dilution and contamination by these proteins. We established platelet

and erythrocyte panels from pure samples of 20 individuals. We then showed that they decrease in the spike-in experiment, which establishes that the contamination originates from erythrocytes or platelets. Therefore, these experiments appear to be suitable. Moreover, the same group of proteins is reported in plasma proteomics studies from other groups as well as found in our previously published studies.

In line with the potential activation of platelets, the secreted or shed platelet proteins from activated platelets are likely the reason for the high number of proteins changing between plasma and serum samples. We mention this already in the paragraph about the coagulation marker panel.

Typo error? in the M&M it is stated at the top of p.16 that the blood is centrifuged at 200g for 10 min. Is this 200g or 2000g?

Author's response: We thank the reviewer the in-depth analysis of the method section. We applied 200g for the centrifugation to separate erythrocytes from the platelet containing plasma to harvest platelet-rich plasma. We further centrifuged both fractions – the erythrocyte and the platelet fraction at 2000g.

Standardization of the preanalytical phase is extremely relevant for discovery proteomics in order to assure that the selected candidate biomarkers are promising and not an artefact due to contamination of plasma (or serum). Setting up an EQA-website with the proposed three quality panels is helpful in order to investigate whether processed biobank samples from research institutes do qualify for the intended research and research questions. Establishing much more comprehensive recommendations for preanalytics seems to be needed. Also, more research is needed to understand the effect of all preanalytical variables (including transport, different matrices, temperature, freezing and thawing, ...).

Author's response: We completely agree with the reviewer that even more recommendations for sample handling would be a useful guidance for researcher in the field. Performing the suggested experiments now allows us to make more specific recommendations towards pre-analytical variations that can result in bias of experiments. These are now incorporated into the revised manuscript (see above).

Referee #2:

1. The technical quality is high, the experiments were designed and correctly performed to make the suggested conclusions.
2. The subject is inherently not novel since numerous publications have already suggested quality controls that can be used to assess plasma sample quality. However, an automated interface where proteomics data can be rapidly screened for sample quality is new as is the ability to designate the source of contamination originating from erythrocytes, platelets, the coagulation cascade or a combination of the three.
3. It is difficult to assess how popular this research tool will be. The medical impact ranges from medium to high.
4. The model system is identical to the system under study (human plasma) and therefore adequate.

Author's response: We thank the reviewer for appreciating the experimental design and our efforts on creating a standardized and public available analysis platform. The novelty of the manuscript lies in the markers themselves, the analysis methods and the experiments how we identified and confirmed the panels by spike-in experiments, former internal studies and the extensive literature

analysis. It is also true that some groups have suggested quality control for plasma samples, but for us it seems that these studies did not really result in actionable quality markers nor have they really found their way into recommendations. Indeed, it is hard to estimate how many people will use the online platform, but we hope that assessing the quality marker panels in clinical studies could become a standard in the field, together with the three proposed methods.

Reviewer summary of the research article:

The manuscript entitled "Plasma proteome profiling to detect and avoid sample-related biases in biomarker studies" By Geyer et al., develops 3 quality indices that can be used to assess pre-analytical sample quality when using plasma for protein biomarker discovery. The three quality indices can be used to identify the source of protein contamination in plasma from erythrocytes, platelets and/or coagulation specific proteins. The quality indices were developed by first performing label-free protein quantification on erythrocytes, platelets, platelet rich plasma, plasma, and serum, all of which were obtained from blood samples donated by 10 males and 10 females. To obtain quality indices that would identify contamination of plasma by proteins from erythrocytes or platelets, proteins which displayed a 10 fold higher intensity in erythrocytes or platelets (compared to plasma) and that had a CV < 30% across all individual samples were selected for quality marker panels. This approach identified 27 unique proteins to erythrocytes and 24 unique proteins to platelets. Subsequently, a set of dilution series was carried ranging from 1:1 erythrocyte/platelet:plasma down to 7 orders-of-magnitude dilution of cells. As expected, this resulted in a decrease of cellular proteins intensity and summing the protein intensities from each panel and dividing each by the total protein intensities in plasma provided two "quality indices" that were linear with respect to cell numbers measured by flow cytometry.

Similarly, to obtain a panel of proteins that could be used to assess systemic bias due to partial coagulation events, 72 plasma and 72 serum samples from 4 individuals were compared via label-free quantitative proteomics. The 30 most significantly altered proteins between serum and plasma were selected, including platelet factor 4 variant 1 because it is an indicator of coagulation. Similar to the aforementioned quality indices, the total protein intensity in plasma was divided by the sum of protein intensities of the 31 protein panel to obtain a coagulation specific quality index which could be used to clearly differentiate plasma from serum. Using these quality indices, the authors then assessed the quality of plasma samples used to perform a previous study in 2016 (Mol Syst Biol 12: 901). Across all 318 plasma samples, the ratio for each quality index was calculated and a ratio higher than 2 standard deviations was used as a cut-off to identify plasma samples that had not been prepared appropriately. Using the protein panels revealed that a number of the plasma samples had been contaminated. The authors also performed an extensive literature search of studies which used plasma for protein biomarker discovery and re-examined the datasets from 210 studies. This approach revealed 113 studies have reported at least one potential quality marker as a biomarker candidate. Based on these results, the authors provide recommendations for sample preparation and make the analysis readily available by creating an online platform at www.plasmaproteomeprofiling.org. Lists of protein intensities can be uploaded which automatically generates the three quality index values, systemic bias and a global correlation map.

Reviewer recommendation:

The developed website which streamlines assessment of pre-analytical plasma sample quality will aid researchers who use plasma for protein biomarker discovery. The automated interface is new, as is being able to define whether the contamination is from erythrocytes, platelets, or incomplete coagulation, or a combination of the three. While other studies make suggestions that quality controls should be included, the present study addresses the subject in greater depth. The recommendation is to publish with the following minor modifications:

Author's response: We thank the reviewer for this encouraging feedback and the detailed summary of our efforts. We also agree with all the minor modifications listed below and included these changes in the revised manuscript. We also added the outline of the template as a supplemental table.

-Page 3 on the last line change "low abundant" to "low concentration" or "low-abundance"

-Page 4 the last paragraph could be improved for clarity. The second line "several hundred plasma samples" I assume this is referring to the 2016b study by Geyer? It is not clear which plasma samples are being discussed as they are not prepared in the present study. This should be explicitly stated. The same is true for the "the study plasma samples" in the same paragraph on line four.
-The figure legend in supplemental Fig. S3 is not properly labeled; panels A and B are not designated in the legend.

-Page 5 "Spiked-in experiments" in the heading, is this valid terminology? Is it not more accurate to say serial dilution experiments validate the erythrocyte and platelet quality marker panels?

-Fig. 2 in the legend could you please state how the Z-score was calculated, it is not obvious.
-Page 7 in the second paragraph "Interestingly, the strongest elevated proteins in serum were all connected to platelets:" It is not clear what is meant by "connected to platelets" are they proteins unique to platelets?

-Page 7 second paragraph "In total, 208 proteins increased and 91 decreased due to coagulation." In plasma or in serum? this should be explicitly communicated here.

-Figure 4A, the y axes is labelled as contamination index but the figure legend and throughout the text these values are described as quality indexes. What is the difference?
-I attempted to use the online platform at www.plasmaproteomeprofiling.org. However, the instructions were not clear on how the text file should be formatted, and I did not succeed in using the platform. I think an example of the formatting or even a template would be helpful.

-Page 16 this sentence should be the starting sentence in the first paragraph, not the second as currently structured. "All participants gave written informed consent for their participation in the Munich Study on Biomarker Reference Values (MyRef), which is registered under the local ethic number 11-16."

-Page 16 please state how much blood was collected from each individual.

Referee #3 (Comments on Novelty/Model System for Author):

No model systems used.

Referee #3 (Remarks for Author):

EMM-2019-10427

Plasma proteome profiling to detect and avoid sample-related biases in biomarker studies
Philipp Geyer et al.

Geyer et al. present a new resource to identify contaminant proteins in human plasma proteomics studies. By acquiring deep reference proteomes, Geyer et al construct quality marker panels for

contamination of erythrocytes, contamination of platelets, and coagulation. The study is of clear interest to the medical and biomarker communities. The manuscript is well written and generally easy to follow, and the authors have provided an online resource to assist the community with screening their plasma proteomes for contaminant proteins. I believe there are only a few issues which should be addressed (as described below).

Author's response:

We thank the reviewer for this positive evaluation and the appreciation of our efforts. We addressed the comments below and made corresponding changes in the revised manuscript.

Major issues:

- Why are there so many proteins with platelet / plasma and erythrocyte / plasma ratios of 10^7 in Fig. 1D and 1E? Please explain.

We thank the reviewer for spotting this. All proteins that were only quantified in erythrocytes or platelets, but not in plasma were aligned at the right side of the graph. We clarified this now in the figures and in the figure legend.

- It is unclear how we should interpret data from the fibrinogens (FGA, FGB, FGG). These proteins are included in the coagulation panel (Fig. 3B), and their disappearance should indicate coagulation. Yet, the authors appear to conclude that the fibrinogens might be biologically relevant in their own biomarker study data because "the fibrinogens group with proteins involved in low-grade inflammation" in the global correlation map (page 9). Does this mean that the fibrinogens are biomarkers of inflammation in this study? If so, how can the authors reconcile this conclusion in their own data with the claim that in many other biomarker studies that fibrinogens are frequently misidentified as biomarkers (page 12)?

Author's response:

Indeed this is not trivial to judge, if proteins are changing due to quality issues or as a real effect from the investigated study. If a panel of quality marker proteins such as the fibrinogens correlate to inflammation markers, then we would argue that there is an overall effect of inflammation on the fibrinogen levels for this special case, especially as the connection between inflammation and fibrinogens is already well known. Nevertheless, if the fibrinogens are regulated together in single samples, this strongly indicates that these samples have quality issues. As we do acknowledge these issues, we try not to judge the published literature. In particular, we write that a number of studies report proteins from the marker panels but not that these are definitely not biomarkers. Moreover, we clearly state in the discussion that the investigated disease or study might have an influence on the different proteins of the panels. For example, that there are diseases that have an influence on erythrocyte lysis or result in increased platelet counts.

- The authors claim on page 9 that "The other two quality marker panels are also readily apparent in the global correlation map", but I do not see the coagulation panel in either Fig. 4C or Fig. S5. The authors should highlight the coagulation panel as well in one of these figures.

Author's response:

The three fibrinogen chains as part of the coagulation marker panel are clustered together in Figure. S5, which we highlighted already.

- I could not find Supplementary Table S1.

- In general, the supplementary tables are not readable because they are spread out over many pages. Several pages in the manuscript pdf are blank because of this problem. The authors should provide this supplementary table in a more readable format, perhaps as an Excel file.

Author's response:

The pdfs of the Excel files were indeed not readable in the combined pdf. The tables including Supplemental Table 1 are now supplied as Excel files.

Minor issues:

- page 4, glyceraldehyde 3-phosphate dehydrogenase has gene symbol "GAPDH" not "GPDH". It is correct in Supp. Table S2.

- page 2, 2nd paragraph, should be "Within its limit of detection" not "Within its' limit of detection" (no apostrophe).

- page 4, "their quantities were not correlate with each other" should be "their quantities were not correlated with each other".

- page 11, "to survey the extent to which probably quality marker proteins are reported...", is the word "probably" necessary?

Author's response:

We thank the reviewer for having spotted these points and included these changes in the revised manuscript.

2nd Editorial Decision

19th August 2019

Thank you for the submission of your revised manuscript to EMBO Molecular Medicine. We have now received the enclosed report from the referees who were asked to re-assess it. As you will see the referees are now overall supportive and I am pleased to inform you that we will be able to accept your manuscript pending the following amendments:

Please address referee #1's comment (Introduction and Quality marker panel for blood coagulation) and referee #3 's comment regarding fibrinogen.

REFEREE REPORTS

Referee #1 (Remarks for Author):

The authors have adequately addressed the remarks of the reviewer.
The extreme relevance of the preanalytical phase has been sufficiently evaluated with the additional experiments.
The other comments are also well addressed.

Minor comments:

- a/ Introduction: first paragraph: cardiac troponins indicating myocardial necrosis (not myocardial infarction);
- b/ Quality marker panel for blood coagulation: first paragraph: Prompt and repeated INVERSION mixes the anticoagulant with the blood (use inversion and not shaking).

Referee #2 (Comments on Novelty/Model System for Author):

The authors have adequately addressed my evaluation.

Referee #3 (Remarks for Author):

The authors have satisfied my previous concerns with the following minor exceptions:

- I now believe that I understand how the fibrinogens can serve as both members of a quality marker panel (coagulation) and genuine biomarkers of inflammation in their previous study. In particular, the authors have a nice statement in the Discussion pertaining to this concept:
"Furthermore, correlation analysis reveals whether or not potential biomarkers emerging from a given study are likely to be associated with quality-related proteome changes instead. Conversely this procedure can 'rescue' genuine biomarker candidates that are part of the quality marker proteomes."

To clarify to the reader that the fibrinogens are an example of this "rescue", the authors should make an explicit mention of the fibrinogens as an example here.

- Supplemental Tables: In their rebuttal, the authors said "The tables including Supplemental Table 1 are now supplied as Excel files.", but no Excel file was provided with the submission. Table S1 is still present as a pdf in the version I was provided for review.

2nd Revision - authors' response

26th August. 2019

We are happy for the positive evaluation of our efforts and include the last required changes in the manuscript.

Corresponding Author Name: Matthias Mann

Manuscript Number: EMM-2019-10427